# An AI-based approach to predict delivery outcome based on measurable factors of pregnant mothers

**Michael Owusu-Adjei** *, **James Ben Hayfron-Acquah, Twum Frimpong, Abdul-Salaam Gaddafi**

Department of Computer Science, Kwame Nkrumah University of Science and Technology, Kumasi, Ghana

* mowusuadjei@st.knust.edu.gh

**Data Availability Statement:** Dataset used for analysis is available through https://github.com/

## Abstract

The desire for safer delivery mode that preserves the lives of both mother and child with minimal or no complications before, during and after childbirth is the wish for every expectant mother and their families. However, the choice for any particular delivery mode is supposedly influenced by a number of factors that leads to the ultimate decision of choice. Some of the factors identified include maternal birth history, maternal and child health conditions prevailing before and during labor onset. Predictive modeling has been used extensively to determine important contributory factors or artifacts influencing delivery choice in related research studies. However, missing among a myriad of features used in various research studies for this determination is maternal history of spontaneous, threatened and inevitable abortion(s). How its inclusion impacts delivery outcome has not been covered in extensive research work. This research work therefore takes measurable maternal features that include real time information on administered partographs to predict delivery outcome. This is achieved by adopting effective feature selection technique to estimate variable relationships with the target variable. Three supervised learning techniques are used and evaluated for performance. Prediction accuracy score of area under the curve obtained show Gradient Boosting classifier achieved 91% accuracy, Logistic Regression 93% and Random Forest 91%. Balanced accuracy score obtained for these techniques were; Gradient Boosting 82.73%, Logistic Regression 84.62% and Random Forest 83.02%. Correlation statistic for variable independence among input variables showed that delivery outcome type as an output is associated with fetal gestational age and the progress of maternal cervix dilatation during labor onset.

## Author summary

This study sample of 842 participants with varying characteristics of pregnancy considered the impact of abortion(s) and fetal deaths on delivery outcome decisions. Results from performing feature selection for variable importance showed fetal gestational age and progress of cervical dilatation to be significant predictors of delivery outcome.

owusuadjeim/maternaldataset/blob/main/mat_measuring_metrics.csv.

**Funding:** The author(s) received no specific funding for this work.

**Competing interests:** The authors have declared that no competing interests exist.

Machine learning modeling of maternal interactions provides additional understanding of what is required for a successful delivery outcome and an appropriate delivery mode that is based on sound clinical judgment which takes into consideration the objective of preserving the lives of both mother and child.

## Introduction

Expectant mothers irrespective of race, geographical location, social status, economic circumstance or ethnic orientation have an inalienable right to a determination of choice of childbirth delivery mode. However, the right to the determination of any particular choice is largely dependent on outcome assessment of balance of risks in individual circumstances together with eventual benefits that is to be derived from the choice made [1]. This is particularly important because, for the multiparous, a previous experience backed by delivery process history is a strong indication for a determined delivery choice. But for the nulliparous, overcoming the fear of uncertainties of any particular delivery outcome poses a greater challenge. Dimensions to delivery mode choice among expectant mothers vary from one to another. To the nulliparous with recurrent spontaneous abortions, the determination of delivery mode for a successful pregnancy (full term pregnancy) is of prime importance due to heightened expectations and increased anxiety [2]. This anxiety among nulliparous [3] often leads to requests for delivery mode choice as compared to the nulliparous with no history of spontaneous abortions. Similarly, to the multiparous with particular delivery mode(s), request for any particular delivery mode is informed by previous delivery experience and current medical conditions of both mother and child. But to the attending medical personnel, child birth delivery remains an outcome of a series of processes, assessments, procedures and evaluations based upon which the final determination of the appropriate delivery form is recommended. This is informed by the desire to reduce and adequately manage pregnancy related complications before and during labor with the ultimate goal of preserving the lives of both mother and child. Additional dimensions to childbirth delivery choice include religious and cultural belief considerations and cultural practices of expectant mothers and their families. These factors also influence the choice for any particular childbirth delivery type(s)[4]. Further dimensions include level of care and quality of interactions between healthcare providers and expectant mothers. In this regard, World Health Organization's (WHO) guidelines on intrapartum care for a positive childbirth experience enumerate certain challenges that must be overcome. One such challenge identified include medicalization of childbirth processes that undermine one's ability to give birth. Labor interventions beyond the reach of many increases health equity gap thereby negatively impacting childbirth experience [5]. Maternal care experiences and expectations, interactions with healthcare personnel, labor interventions and medicalization processes in childbirth delivery is summarized in a graphical presentation shown in the interactive flow-chart diagram in Fig 1.

## Maternal interactions

Maternal interaction among entities during the process of childbirth as shown in Fig 1 illustrates interactive components at play during delivery processes. Experiences with these components form the basis for any particular conclusions drawn by expectant mothers on delivery expectations (positive or negative). Positive feedback from patient involvement in critical decision-making serves as an important endpoint for women in labor. This feedback reflects fulfilment of personal expectations including religious, cultural and socio-economic beliefs and

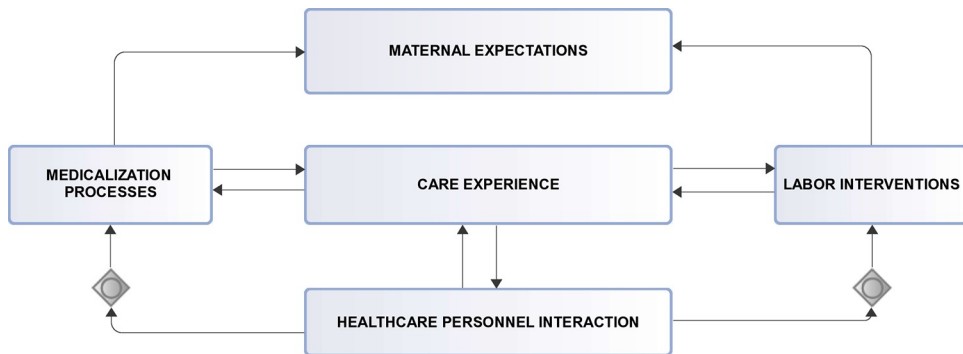

**Fig 1. Maternal interactions flowchart.** Caption: Maternal care experiences and expectations, interactions with healthcare personnel, labor interventions and medicalization processes on childbirth delivery.

practices [5]. To achieve this positive feedback from interventions and personnel interactions, various research studies have examined individual levels of interactions and how these interactivities affect childbirth delivery outcome. A cohort study that compared long-term reproductive and obstetric outcomes in women treated for fear of childbirth and those without this fear for all nulliparous [6] concluded that fear of childbirth among this group of women persists even in subsequent pregnancies. Additionally, the prevalence and impact of fear of childbirth and associated risk factors in another research study [7] reported incidence of fear of childbirth among nulliparous women. A focus study on labor interventions to address the fear of childbirth in order to reduce negative birth experiences [8] proposed an evaluation of counseling interventions in clinical practice. Hypothetical assessment of outcome status on methods of delivery with mothers age [9] concluded that delivery outcome is independent of the maternal age. Maternal state (nulliparous or multiparous) is identified to be associated with delivery mode among other studies. Interaction related studies to examine the role of healthcare personnel on effectiveness of labor interventions and medication processes that meet maternal expectations have also been undertaken. Delivery outcome expectations among pregnancy types remain varied and this is significantly due to differences in pregnancy circumstances. For the nulliparous with few or recurrent spontaneous abortions, childbirth delivery anxiety remains a greater challenge. It is therefore important to describe the various pregnancy types in any meaningful estimation of childbirth delivery mode for better and proper evaluation. Additionally, risk assessments to determine factors that influence delivery type may also consider the number of successful pregnancies, number of spontaneous abortions for both multiparous and nulliparous, number of fetal deaths including still births, maternal age, gestational age and other measurable metrics with direct consequences on delivery outcome. Progress of maternal cervix dilatation over a given time interval, maternal blood pressure, fetal heart rate, maternal haemoglobin count, fetal weight and maternal temperature together with amniotic fluid index among others must also be considered in delivery risk assessments. This research work focuses on the use of artificial intelligence-based techniques to explore the impact of using measurable metrics on delivery outcome. Features included in this work are maternal blood pressure at the onset of labor, fetal heart rate, maternal pulse rate, maternal haemoglobin count, maternal cervical dilatation count at the onset of labor over three hours interval, gestational age, fetal weight, number of successful pregnancies, abortions and fetal deaths. To estimate output variable independence, a unique feature selection technique called Chi-square correlation statistic test, is performed to help discover relationships between measurements of these characteristics and is impact on potential delivery outcome.

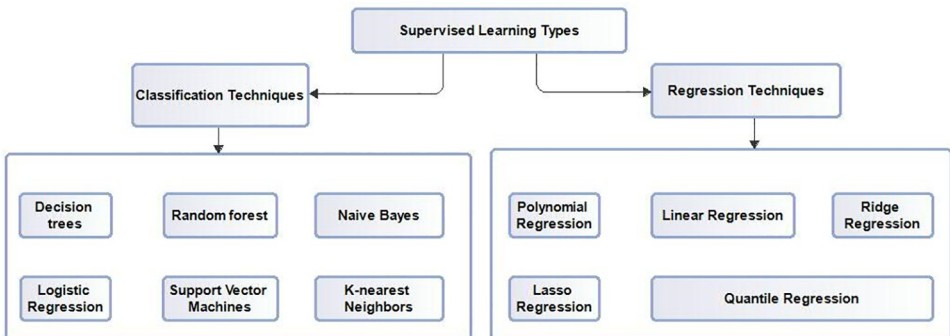

**Fig 2. Supervised machine learning types.** Caption: Supervised learning types and the various algorithms they represent.

## Supervised machine learning techniques and types

Supervised learning is one in which labeled examples of data (input and output) are used to train an algorithm to correctly predict or classify an output label from the input data. It is predominantly useful in real-world applications such as fraud detection, spam filtering applications, disease diagnosis, financial risk assessments, sentiment analysis, image and document classifications, etc. In data mining, supervised learning is used in two problem domains: output classification, where the output variable is categorical, such as true or false, yes or no, etc and regression, where relationships between dependent and independent variables are investigated for impact on the output variable. Regression enable us to answer the question of variable importance in predictive analysis. Supervised learning algorithm task is to find an appropriate mapping function to map an input variable (independent variable) (x) into an output variable (dependent variable) (y). Supervised learning types and the various algorithms they represent is simplified in graphical representations shown in Fig 2. Supervised learning techniques can be grouped into two main categories based on its use; Regression and Classification.

## Related research works

Previous research studies on interactions with healthcare personnel, impact on interventions, detection or identification of variable importance and other relevant works to address patient expectations is also focused on patient characteristics that enhances delivery and other treatment outcomes [10]. An exploratory study to understand women's expectations with respect to personnel interactions during labor explored the following themes: provider match, safety/ risks, decision making and care satisfaction [11]. For those with expectations that matched provider services, they expressed positive experiences; ironically, on interventions, there are mixed feelings about their use and appropriateness, attributed to bureaucratic and complicated processes. The expression of mixed feelings and growing concerns about commonly used childbirth interventions (induction of labor, Augmentation of labor, Artificial rupture of membranes, Episiotomy in vaginal births, Caesarean section etc) is echoed in another study [12]. This study considered the impact of variations in childbirth interventions in high-income countries for multiparous and nulliparous women. The importance of interventions and its impact is underscored. But its use routinely in healthy women is estimated to cause maternal and neonatal harm hence the challenge to address ideal rates of use of interventions. Additional interactions which involves family members (spouses, partners, family members or friends) accompanying expectant mothers as companion of choice during labor is also

estimated to improve childbirth delivery outcomes [13]. One identifying setback in this interaction is its implementation as many healthcare facilities in most countries lack clear policy direction on its use. Perceiving policy decisions as a challenge for reddress can lead to improvements in childbirth delivery outcomes. Childbirth delivery interventions are implemented to improve delivery outcomes, but one great challenge to its implementation is the required skill and knowledge by healthcare managers to implement these interventions and how they can impact on important clinical decisions [14]. This study concludes that critical thinking skills and appropriate communication skills were important ingredients necessary for clinical decision. However, challenges in variations of level of knowledge exihibited by primary care midwives influencing clinical decisions on childbirth delivery intervention use were identified. Further, statistical evaluation studies [15] to establish or determine factors influencing maternal decision for a choice between caesarean section and vaginal delivery identified relationships between three factors: culture, lifestyle and perception as the most important variables in the decision for a choice between caesarean section and normal delivery by mothers.

Common expectations among every expectant mother and attending healthcare personnel is a process that guarantees safe delivery with reduced risk of pain and other related complications especially in the use of medicalization and labor interventions. In this endeavor, dealing with the psychological effects resulting from the use of interventions such as (episiotomies, forceps or or vacuum extraction, C-sections, induction, etc.) could also address maternal expectations and lead to a more positive childbirth experience [16]. In view of the psychological and other negative effects associated with labor interventions, a recent focus on childbirth delivery with minimal interventions has emerged. The desire to limit intervention use is a shared concern by both healthcare providers and expectant mothers [17]. If this desire is to be achieved, then the need to identify patterns of change that necessitate these interventions is of utmost importance. Predictive machine learning approach has the potential to identify these patterns of change. The prediction of pre-maturity from medical images in the review of perinatal complications with support vector machines yielded an accuracy score of 95.7% and the prediction of neonatal mortality with XGBoost technique produced an accuracy score of 99.7%[18]. Further studies to predict mode of delivery using Support vector machines, Multilayer Perceptron, and Random Forest techniques to develop clinical decision support systems for the prediction of mode of delivery specific to three categories: caesarean section, euthocic vaginal delivery and, instrumental vaginal delivery recommended limits. With an estimated sample population of 25,038 records consisting of 48 attributes, [19] using women with singleton pregnancies, the performance of three algorithms were similar with 90% classification accuracy score for caesarean section and vaginal delivery and 87% between instrumental and euthocic delivery types. High caesarean section rates as reported [19] is echoed in a related article with the objective of determining sub-types of women at higher risk of caesarean section delivery [20] by using demographic, clinical and organizational variables with classification tree analysis. Conclusions drawn indicate that clinical variables are important predictors of caesarean delivery.

Further, predictive modeling [21] of emergency cesarean section with logistic regression, random forest, support vector machine (SVM), gradient boosting, extreme gradient boosting (XGBoost), light gradient boosting machine (LGBM), k-nearest neighbors (KNN), voting, and stacking showed that using variables such as maternal age, height, weight at pre-pregnancy, pregnancy-induced hypertension, gestational age, and other ultrasound findings about the fetus showed logistic regression accuracy score of 78%. Clinical and Sonographic findings obtained at term are identified to be best predictors of emergency caesarean section need. An assessment of the possibility of vaginal delivery after a caesarean section [22] found limitations in the implementation of calculators into clinical practice. The study therefore centered on assessing the feasibility of machine learning models in addressing these limitations. Study

conclusions showed that applying machine learning algorithms that assigns individual risk score for every successful vaginal delivery after caesarean section may assist in future decision making for delivery outcome determination.

Pre-term deliveries are a worldwide health concern especially to expectant mothers and their immediate families due to associated complications in its management and the attendant deaths resulting from these complications. Predictive algorithm use with improved accuracy based on important variable features is a challenge that must be overcome. The adoption of entropy feature selection strategy is viewed as means of overcoming this challenge [23]. Using three classifiers namely; decision tree (DT), logistic regression (LR), and support vector machine (SVM), SVM generates a prediction accuracy score of 90.9% as the highest accuracy rate. Using an inclusion and exclusion criteria in the sampling process which included gestation age of 28 weeks or older, women who delivered live births, registered with antenatal clinic attendance. Exclusion criteria were; women with multiple gestation (twin gestation), women with stillbirths and women referred to other hospitals.

In its establishment [24] of labor risk scores for maternal and neonatal unfavourable delivery outcomes using machine learning techniques, dataset characteristics of mean gestational age 39.35 ± 1.13 weeks, mean maternal age 26.95 ± 6.48 years and mean parity of 0.92 ± 1.23 are used. This study achieved different accuracy scores at different cervical dilatations. At a cervical dilatation of 4 centimeters (4cm) an accuracy score of 75% was achieved and at cervical dilatation of 10cm, 89% accuracy score was achieved. A systematic review [25] of pregnancy outcomes with machine learning for optimal delivery mode, showed that the use of unsupervised learning techniques together with deep learning algorithms for prediction, results in the determination of reasons for maternal complications previously unknown.

In the wake of increasing use of artificial intelligence and predictive techniques in various fields and by extension to the healthcare system, ethical considerations regarding data generation, use and acquisition mechanisms [26] has become critically important. The increasing use of artificial intelligence and machine learning in healthcare applications is also underscored in a related study [27] which considered reliable prediction model for maternal care decision support systems based on data collected on antenatal signs and symptoms (enriched data) to predict mode of childbirth delivery before term. Conclusions in this study suggests that the use of "enriched data" contributed to high model performance in sensitivity, specificity, F1-score and receiver operating characteristic curve score (auc). Prediction accuracy scores achieved by the various learning techniques; k-nearest neighbor was 84.38%, bagging was 83.75%, random forest was 83.13%, decision tree was 81.25%, and AdaBoostM1 was 80.63%.

However, a study to determine the effect of socio-demographic effects on caesarean section delivery [28] identified closed relationships between a womans level of education, income level, habitat and health conditions such as hypertension for both primiparous and multiparous women.

## Summary of related works

Many of the concepts identified in the related research work can be linked to maternal interaction diagram in Fig 1. Concepts involving labor interventions such as identifying predictors of childbirth delivery, healthcare personnel interactions with patients, knowledge and skills of healthcare personnel in administering labor interventions, medicalization processes (identified medical conditions) that requires care and support, impact of variations for childbirth intervention for both multiparous and nulliparous women have been examined in various studies.

Problems identified in related works include prediction of vaginal delivery outcome after a caesarean section [28], estimating labor risk score for maternal and neonatal delivery outcome

[24], systematic review [25] on pregnancy outcome with machine learning for optimal delivery mode, adoption of entropy feature selection strategy as means of overcoming challenges of use of important variable features [23], determining sub-types of women at higher risk of caesarean section delivery [20], predictive modeling [21] of emergency cesarean section as a delivery outcome, psychological and other negative effects associated with labor interventions [17], identifying pregnancy related complications [17], understanding womens expectation with respect to healthcare personnel interactions during labor [11], impact on variations in childbirth interventions in high-income countries for multiparous and nulliparous women [12] and many others using various features both demographic and medical records of patients.

In determining childbirth delivery outcome for an expectant woman either by healthcare personnel or through maternal or patient request, understanding of pregnancy history related to the number of unsuccessful pregnancies (spontaneous abortions or otherwise), number of stillbirths (fetal deaths if any) in addition to known and unknown factors may provide useful insight into critical clinical decisions and the underlying reasons for which patient request is treated.

It is the non-inclusion of these factors (unsuccessful pregnancies (spontaneous abortions or otherwise), number of stillbirths (fetal deaths if any) that is identified as a research gap that must be addressed. This research therefore, includes these factors in the predictive model for delivery outcome determination to help bring to the fore the impact of these factors on the design of any decision support system for delivery outcome with efficient AI-based technique applications.

## Research hypothesis

*Null hypothesis*: No relationship exists between delivery outcome and measurable metrics taking into account incidence of abortion(s) and fetal deaths that may have occurred.

*Alternate hypothesis*: Taking into account the history of abortion(s) and any fetal deaths that has occurred in the lives of a pregnant woman (spontaneous or otherwise), relationship exists between delivery outcome and real-time measurable metrics obtained from the partograph. To determine the impact of these metrics on delivery outcome, feature selection technique with Chi-square correlation statistic for variable independence is performed for each feature selected to determine its impact on delivery outcome. The determination of best prediction accuracy using balanced accuracy from the evaluation of three (3) machine learning techniques namely; Logistic Regression, Random Forest and Gradient boosting classifiers are used in this context.

## Research materials and methods

Formal request for permission to use healthcare facility for this exercise was made through a correspondence dated 10th December, 2019 referenced Ds24/2019. Correspondence confirming grant of permission was received on 5th January, 2020 referenced KGHR210/2020.

Quantitative research approach involving the use of structured methods for data collection and analysis was adopted. Focus on maternal history to include total number of spontaneous abortions or otherwise, number of stillbirths (fetal deaths if any) to understand how history of previous pregnancy outcomes could influence birth delivery outcome is emphasized. This could lead to a better understanding of contextual parameters with significant contributions to childbirth delivery type phenomenon. To achieve this objective, purposive sampling technique for the following category of participants was adopted. These included those who had not given birth before (irrespective of the outcome of previous pregnancies-nulliparous), those in their first pregnancy (primagravida), those who had given birth once (primiparous), those

**Table 1. Sample Population Count (conceptions, deliveries, abortions and fetal deaths).** Statistical distribution of sample population counts.

| Description | Count |
|---|---|
| 1st time pregnancy with no abortion(s) | 186 |
| Patients with abortions | 215 |
| Abortions with no live birth | 47 |
| Abortions with live births | 168 |
| More than one pregnancy, No births but more than (0) abortions | 47 |
| More than one pregnancy, No births | 251 |

who had been pregnant more than once (multigravida), those who had given birth more than once (multiparous), those who had given birth to five or more infants with gestational age of 24 weeks or more (grand multipara), those who had been pregnant for more than five times (grand multigravida) and finally, those with seven or more deliveries beyond 24 weeks of gestation (great grand multipara). Partograph records of 842 patients were purposively sampled from patients who had delivered at Kwahu Government Hospital from January 2020 to September 2020. The only exclusion criteria used was to eliminate partographs that had not been filled properly or had empty spaces (unfilled sections). Partograph [29] is seen as an essential tool for skill management of delivery process, recording labor progress, maternal and child conditions in real-time for decision making. It's the main data source for use together with other medical records in this research work.

The sampled population had a mean gestational age of 38.76±2.00 weeks and a mean patient age of 27.69 ± 6.53 years. Minimum patient age recorded was 14±6.53 years and maximum patient age recorded was 45 ± 6.53 years.

Participants within 23 years of age were at the 25th percentile, which means 25% of all participants were 23 years and below, 50% participants were below and above 28 years (50th percentile) and 75% were younger than 32 years (75th percentile). Total number of sample population between the ages of 23 years and 32 years were 421. The patient with the highest pregnancies had 17 (gravida 17) with 11 deliveries (para 11) at age 40 years. Other exploratory statistics including study sample population characteristics are shown in Table 1.

## Ethical approval and consent

Ethical approval to conduct this research was obtained from The Clinical cordination team of Kwahu Government Hospital for the use of Electronic healthcare record dataset with approval notice referenced KGHR210/2020 on 5th January, 2020. All possible patient/participant/sample identifiers such as names, location addresses and many others were removed to ensure patient/participant privacy and protection.

## Sample attributes

This research work was conducted in a government funded health facility. It has a minimum outpatient attendance of 450–500 patients per day. Post-natal and antenatal services constitute over 50% of total services rendered for an estimated population of over 200,000 inhabitants with varying ethnicities and professions. Inhabitants are predominantly traders and farmers but large segments of the population can be classified as educated elite. This is so because the geographical location is home to several educational institutions and governmental agencies.

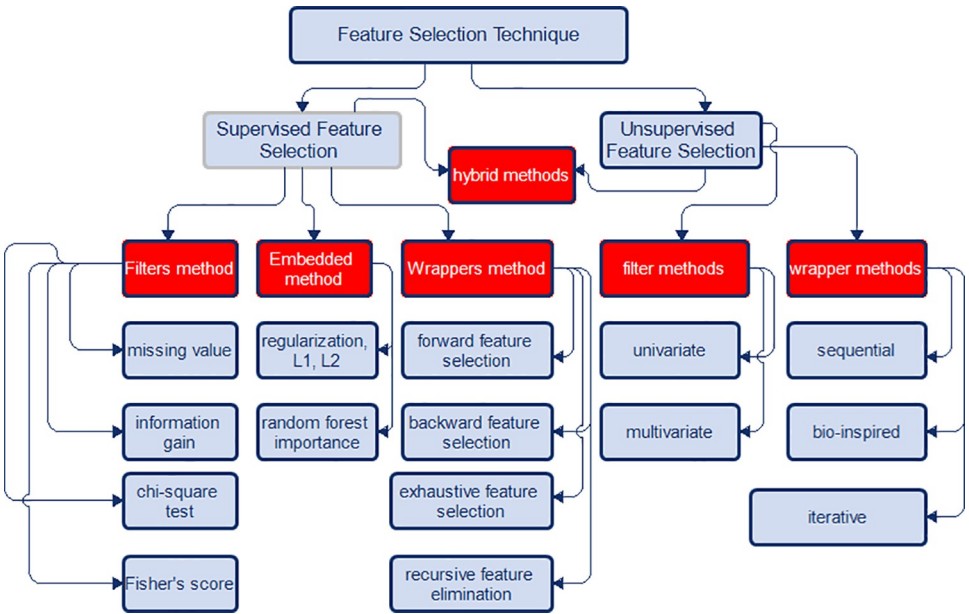

**Fig 3. Feature selection techniques.** Caption: Presentation of feature selection techniques based on four methodologies namely; filter methods, wraper methods, embedded methods and hybrid methods.

### Feature selection and feature categories

Feature selection for relevant feature from collected electronic health records of patient information gathered from ante-natal hospital attendance is achieved with supervised feature selection technique to ensure that redundant, irrelevant and noisy features are excluded. The specific feature selection technique adopted in this research work was chi-square correlation statistics which is part of filters method for supervised feature selection.

### Chi-square test

Chi-square test, a filter feature selection method is used to determine feature relationships between categorical variables. The chi-square value is calculated between each feature and the target variable for which the desired number of features with the best chi-square value and alpha value of less than 0.05 is selected. Presentation of feature selection techniques based on four methodologies namely; filter methods, wraper methods, embedded methods and hybrid methods is illustrated in Fig 3. Among feature categories considered in this research are nulliparous, primagravida, primiparous, multigravida, multiparous, grand multipara, grand multigravida and great grand multipara.

## Results

The study sample of 842 participants with varying characteristics of pregnancy considered the impact of including counts of abortion(s) and fetal deaths on delivery outcome decisions. First time pregnancies without abortion(s) were 186, total patients with the history of abortion(s) were 215 made up of abortion(s) with live births 168, and abortion(s) with no live births 47.251 patients with more than one pregnancy but zero parity (no births) were also recorded. Fig 4 has three (3) boxplot images that describe skewness of data used, (gravida, parity and abortions) to show data distribution or spread. It can be seen that the median value for gravida is 1 shown by the line in the box and one extreme outlier of value. Both parity and abortion

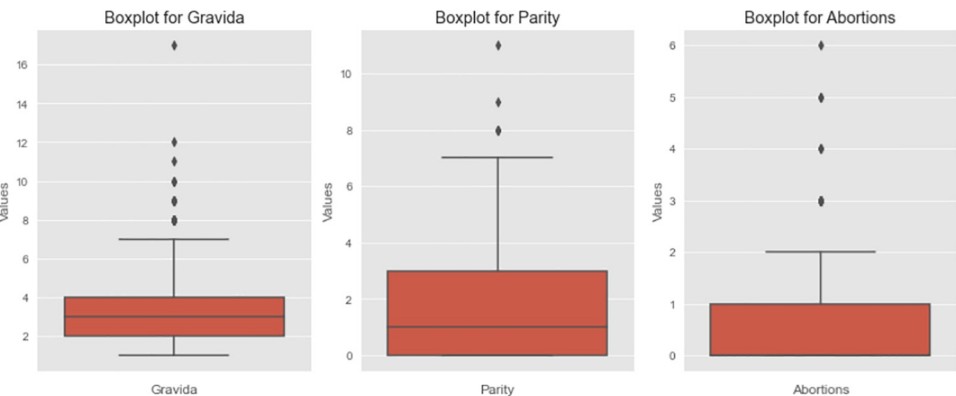

**Fig 4. Boxplot images: Caption.** Boxplot images that describeskewness of data used, (gravida, parity and abortions) to show data distribution or spread.

boxplots have only one whisker, which means either, their minimum values in both instances are equal to that of the lower quartile or the maximum values are equal to the upper quartile. Abortions boxplot has no median line which means that the median value is equal to either the lower or upper quartile. Additional exploratory analysis with scatter to determine relationships in the collected data is also shown in Fig 5. Three features are identified in Fig 5 (gravida, parity and abortions) and shows the density of occurrence in each plot. Recorded abortions under 1 reported were fewer than those between 1 and 3. Number of deliveries (parity) recorded were higher between 0–4 than above 4. Number of pregnancies recorded from1 to 6 were high than above 6. Fig 6 describes the various processes and demonstrates potential areas of data collection for predictive modeling purposes. It also includes sub-processes, which in this instance served as major source of data collection.

Among the objectives of this research work was the determination of variable independence on the prediction outcome, Table 2 describe results obtained from performing Chi-square correlation statistic test with collected features to determine feature relationship with the dependent variable. Two dependent variables are identified as those with correlations to the prediction outcome and these are gestational age (in weeks) and progress of maternal cervical dilatation. Figs 7, 8 and 9 are confusion matrices of the three algorithms used namely; Fig 7 (Logistic Regression), Fig 8 (Gradient Boosting) and Fig 9 (Random Forest) and each contains

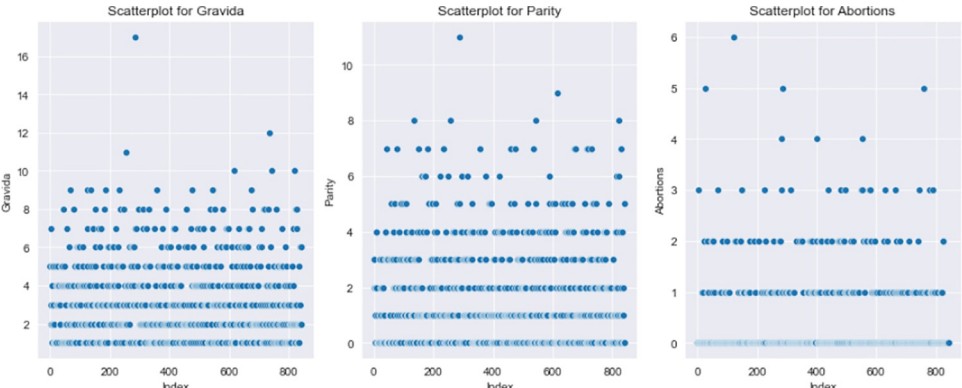

**Fig 5. Scatter plot: Caption.** Scatter plot to determine relationships in the collected data. Three features are identified in the collected data are (gravida, parity and abortions) and shows the density of occurrence in each plot.

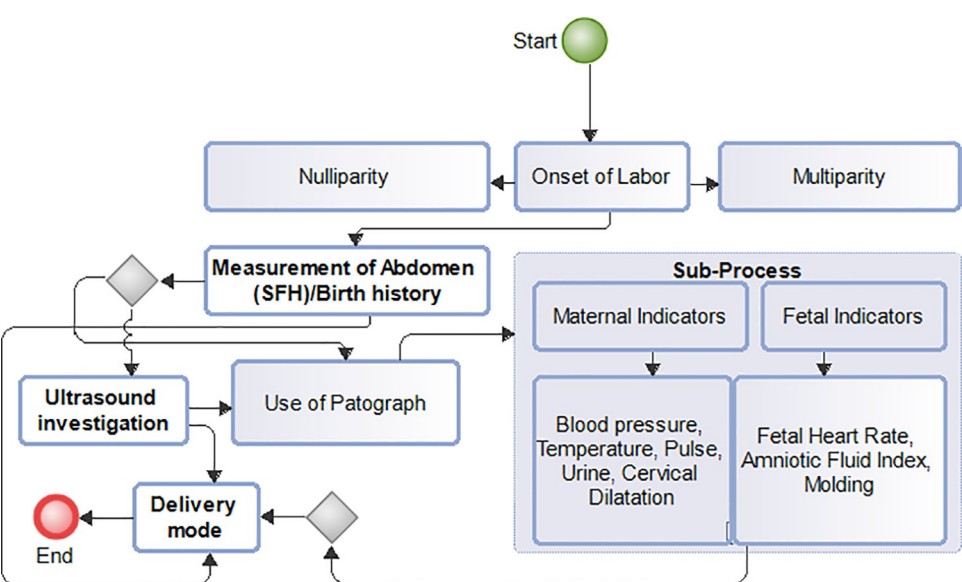

**Fig 6. Areas for Feature collection: Caption.** Describe the various processes and demonstrates potential areas of data collection.

descriptions of predicted values for true positive predictions, false positive predictions, true negatives and false negatives. These are represented as True negatives (TN), True Positives (TP), False Negatives (FN) and False Positives (FP).

In Table 3, predicted probability scores for each algorithm is displayed together with balanced accuracy scores in each instance. One of the key objectives was to evaluate model prediction accuracy performance with balanced accuracy scores as real world applications contain imbalanced datasets for which contributions from the minority class is overlook by the majority class. Using balanced accuracy scores instead of roc_auc scores will help address this challenge. In Fig 10, a display of roc_auc score curve shows the following scores obtained by each

**Table 2. Feature Dependency Statistics.** Correlation Statistic for variable dependence.

| Feature | Chi-square value | p-value | Relationship with output |
|---|---|---|---|
| parity | 8.24 | 0.99 | failed to reject $H_0$ |
| gravida | 14.19 | 0.97 | failed to reject $H_0$ |
| pa_age | 35.64 | 0.99 | failed to reject $H_0$ |
| abortions | 5.68 | 0.97 | failed to reject $H_0$ |
| Fet_deaths | 11.43 | 0.18 | failed to reject $H_0$ |
| Mat_bp_systolic | 140.82 | 0.88 | failed to reject $H_0$ |
| Mat_bp_diastolic | 141.76 | 0.31 | failed to reject $H_0$ |
| mat_temp_ini | 37.12 | 0.60 | failed to reject $H_0$ |
| mat_Hb | 81.37 | 0.99 | failed to reject $H_0$ |
| cerx_dil | 366.93 | 4.43e-63 | rejected $H_0$ |
| ges_age | 73.00 | 0.00 | rejected $H_0$ |
| mat_pul_ini | 111.60 | 0.97 | failed to reject $H_0$ |
| fhr_ini | 70.20 | 0.98 | failed to reject $H_0$ |
| anc_vis | 20.92 | 0.98 | failed to reject $H_0$ |
| f_gender | 2.62 | 0.62 | failed to reject $H_0$ |
| f_weight | 56.94 | 0.98 | failed to reject $H_0$ |

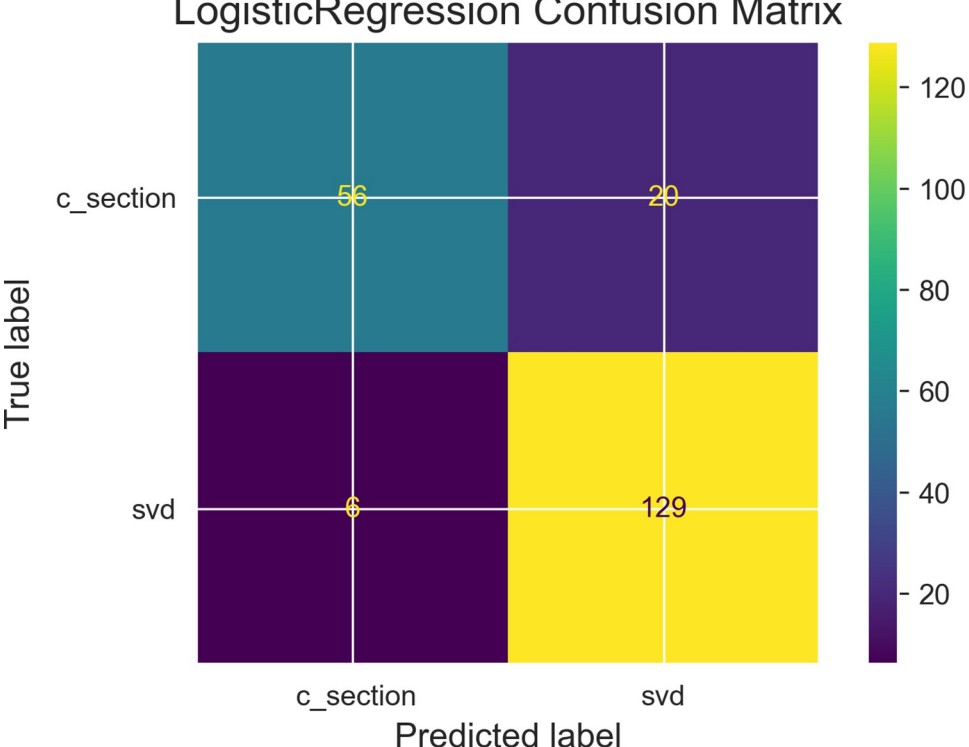

**Fig 7. Logistic regression confusion matrix: Caption.** Contain descriptions of predicted values for true positive predictions, false positive predictions, true negatives and false negatives.

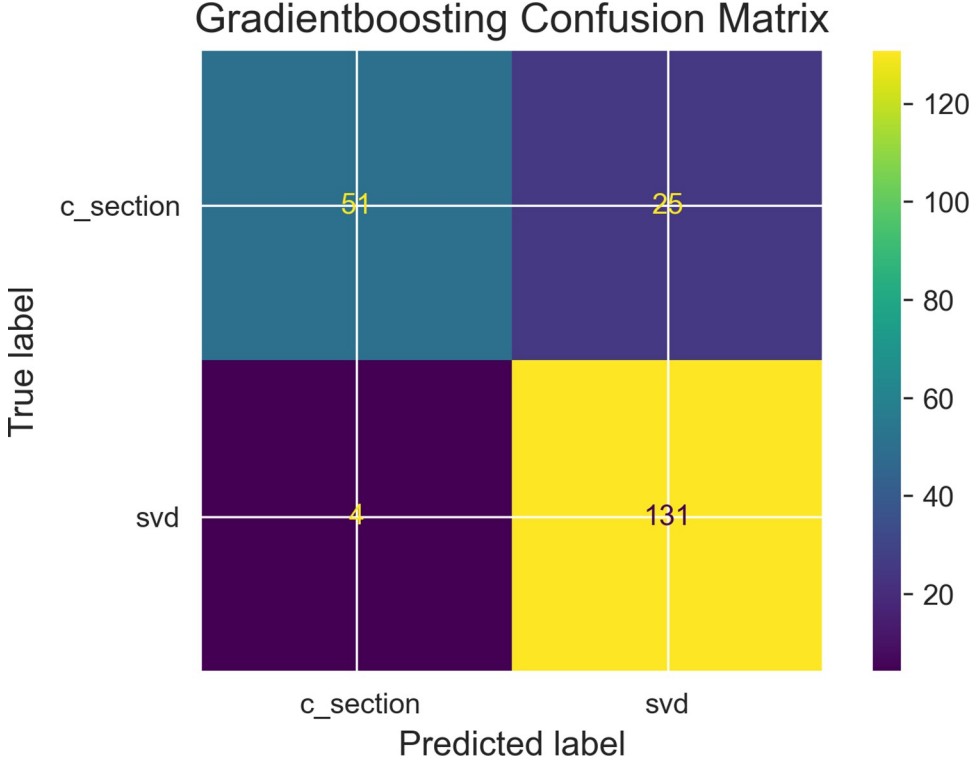

**Fig 8. Gradient boosting classifier confusion matrix: Caption.** Contain description of predicted values for true positive predictions, false positive predictions, true negatives and false negatives.

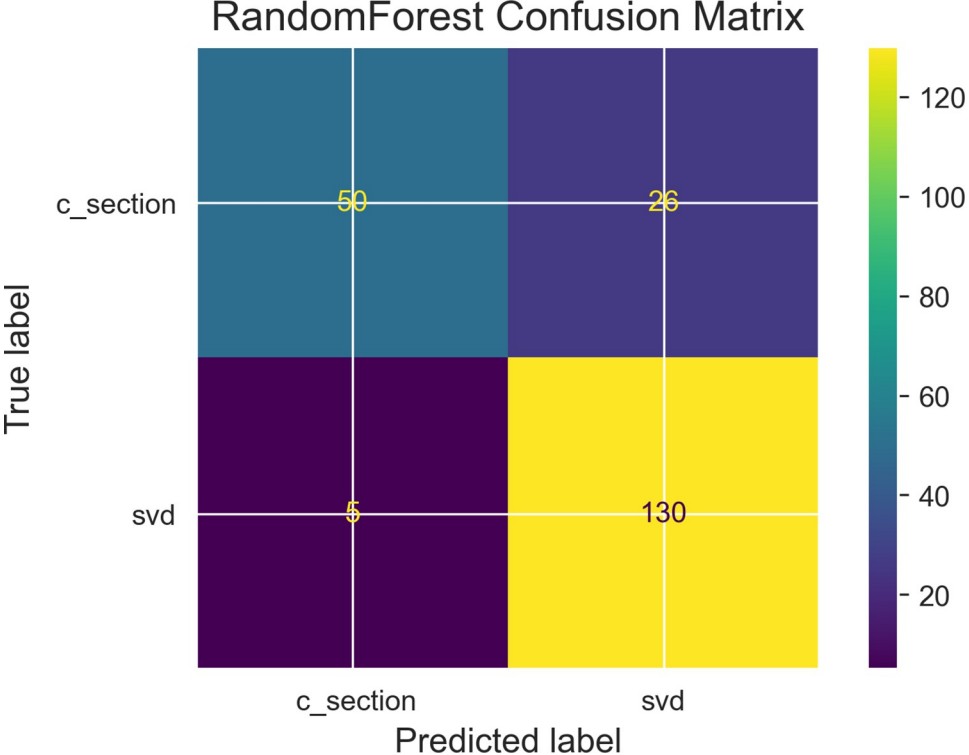

**Fig 9. Random forest classifier confusion matrix: Caption.** Contain description of predicted values for true positive predictions, false positive predictions, true negatives and false negatives.

machine learning algorithm used; Logistic regression has 93%, Random Forest has 91% and Gradient boosting has 91%.

Table 3 is a display of performance scores from model evaluations showing false negative rates (FNR), true negative rates (TNR), false positive rates (FPR), predicted positive values (PPV), negative predicted values (NPV), true positive rates (TPR), f1-score and balanced accuracies for the three models used.

FNR = FN

FN +TP where FN = False Negatives and TP = True Positives

TNR = TN

TN + FP Where TN = True Negatives, FP = False Positives

FPR = FP

FP + TN Where FP = False Positives, TN = True Negatives

PPV = TP

TP + FP Where TP = True Positives, FP = False Positives

NPV = TN

TN + FN Where TN = True Negatives, FN = False Negatives

**Table 3. Model Evaluation Performance Results.** Performance scores.

| Model | FNR | TNR | FPR | PPV | NPV | TPR | F1 | Balanced Accuracy |
|---|---|---|---|---|---|---|---|---|
| Gradient boosting | 2.96% | **68.42%** | 31.58% | 84.52% | 92.86% | 97.04% | 90.34% | **82.73%** |
| Logistic regression | 4.44% | **73.68%** | 26.32% | 86.58% | 90.32% | 95.56% | 90.85% | **84.62%** |
| RandomForest | 3.7% | **69.74%** | 30.26% | 84.97% | 91.38% | 96.30% | 90.28% | **83.02%** |

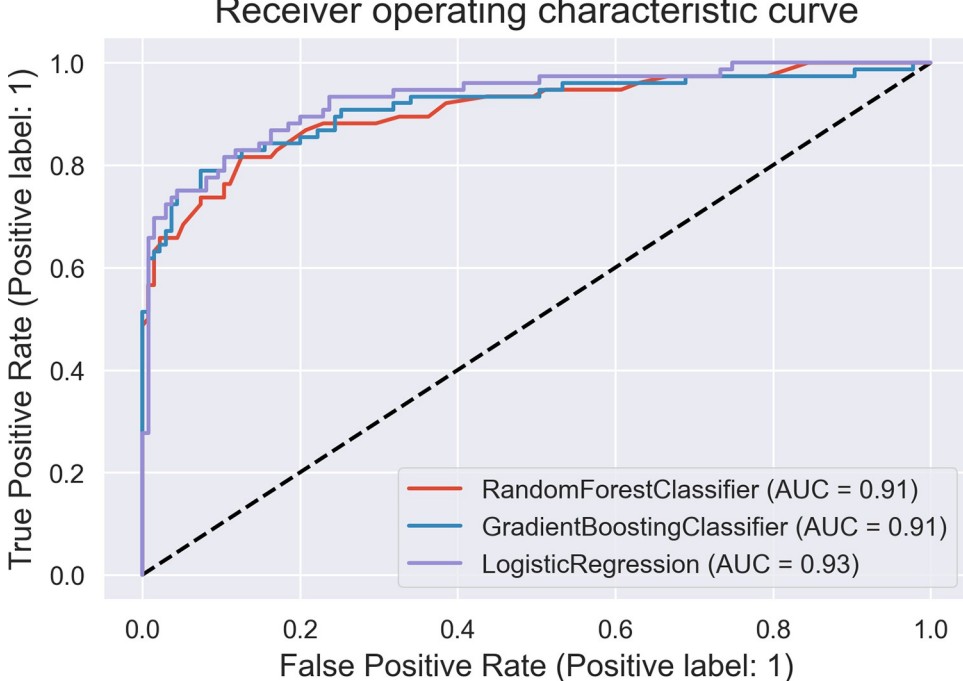

**Fig 10. Receiver operating characteristic curve: Caption.** Display of roc_auc score curve that shows scores obtained by each machine learning algorithm used; Logistic regression has 93%, Random Forest has 91% and Gradient boosting has 91%.

$$TPR = \frac{TP}{TP + FN}$$ Where TP = True Positives, FN = False Negatives

Display of (probability) predicted scores aggregated at threshold points referred to as the auc score by the individual machine learning techniques is shown in Fig 10. Prediction accuracy scores of each model is indicated as Random Forest 0.91, Gradient Boosting Classifier 0.91 and Logistic Regression 0.93.

## Discussion

This section begins with emphasis on whether research objectives have been achieved. One of the key components set out in the research hypothesis was a determination of impact of used features taking into account the inclusion of abortion(s) and fetal deaths. Results from performing feature selection for variable importance with Chi-square correlation statistic as shown in Table 2 indicate p-values for abortions and fetal deaths as 0.974 and 0.179 respectively which are greater than p-value of alpha (0.05 confidence interval) therefore the test failed to reject $H_0$ in both instances. However, p-values for two other features (fetal gestational age and progress of cervical dilatation) showed statistical significance with P-value of alpha for fetal gestational age obtained was **0.00** and for cervical dilatations p-value of alpha was **4.43e-63**. These results have been highlighted for emphasis. The two features therefore rejected $H_0$ indicating correlation in delivery outcome. They prove to be significant predictors of delivery outcome. They were also identified as significant predictors of post partum hemorrhage in a risk prediction modeling research [30]. Fetal gestational age as a predictor of delivery outcome is also shared in a related study for the prediction of labor outcome [31] which among other factors mentioned gestational weeks of 37 and 38 as significant correlated variables to delivery outcome. Gestational age as a significant factor is underscored in other related works such as

by [32]. Among factors predicting vaginal delivery as an output after labor induction, gestational age $\leq$ 39 weeks is listed as a significant predictor by [33]. Progress of Cervical dilatation as a predictor of delivery outcome is also underscore in a related study that used advanced cervical dilation as a predictor for low emergency caesarean section delivery [34]. The use of both features (fetal gestational age and cervical dilatation) as predictors in various research studies therefore gives credence to the predictive capabilities of the factors involved. Evaluation metrics such as area under the receiver operating characteristic curve (auc_roc) and prediction accuracy score shown in Fig 10 (roc_auc score graph) and balanced accuracy score shown in Table 3, indicate high performing traditional machine learning models. An roc_auc score of 91% obtained with random forest is comparable and even higher than results obtained in similar research settings [35] score of 86%. The justification for its use is also underscored in similar research findings [36] conducted to predict the risk of birth asphyxia and in the prediction of intrauterine growth restriction which used deep learning techniques and obtained an roc_auc score of 91% [37].

Model evaluation performance from Table 3 show predicted probabilities for True Negative Rates as; Logistic Regression: 73.68%, Gradient boosting: 68.42% and Random Forest: 69.74%. Prediction of an outcome (positive-csection, negative-svd:1, 0) for each algorithms probability accuracy is determined by its percentage. Prediction of svd as a delivery outcome by logistic regression will be 73.68% accurate, 68.42% accurate for Gradient boosting and 69.74% accurate for Random forest therefore Logistic regression has a lower prediction error score than the other two algorithms. This makes Logistic Regression the algorithm of choice. Machine learning modeling of interactions as shown in the maternal interactions flowchart in Fig 1 and the delivery process flowchart in Fig 6 provides a clearer and better understanding of what is required for a successful delivery outcome and an appropriate delivery mode which is based on sound clinical judgment that takes into account the objective of pre-serving the lives.

## Research contribution

In this research work, patient's history of previous abortion(s) and fetal deaths have been added to already known variables predominantly used in determining childbirth delivery outcomes in known related research works. Prediction scores obtained with roc_auc for these traditional modeling techniques such as random forest are comparable and in this instance competitively higher than those obtained with advanced techniques as stated in the discussions section. Graphical display of maternal interaction flowchart diagram in this work simplifies childbirth delivery process for enhanced understanding. Real-World applications such is in medical fields have unequal dataset class distribution (imbalanced dataset) problems therefore model evaluation metrics used for performance assessment may take into account minority class contributions. The disparity in output class distributions is discounted by most machine learning techniques giving an erroneous impression of a relatively high prediction accuracy score performance (if prediction accuracy is the focus) in such studies. The use of balanced accuracy score obtained from computed predicted true negative values, true positives values, false negative values and false positive values will lead to the determination of best model performance in instances where minority class determination is a major priority such as healthcare systems.

## Strengths and limitations

Strengths identified in this study are two-fold, one is in the determination of feature correlation. Chi-square correlation statistic showed two feature correlations which are subject of related research studies and therefore confirms the validity of our research results. A second

**Table 4. Features used and their descriptions.** Variable feature selections used.

| Feature | Description |
|---|---|
| pa_age | Age of patient |
| gravida | Number of pregnancies |
| parity | Number of deliveries |
| abortions | abortions |
| fet_deaths | Fetal deaths |
| mat_Bp_systolic_ini | Maternal systolic blood pressure |
| mat_Bp_diastolic_ini | Maternal diastolic blood pressure |
| mat_temp_ini | Maternal body temperature |
| mat_Hb | Maternal haemoglobin count |
| cerx_dil | Cervical dilatation |
| ges_age | Fetal gestational age |
| mat_pul_ini | Maternal pulse |
| fhr_ini | Fetal heart rate |
| anc_vis | Number of antenatal visits |
| f_gender | Fetal gender |
| f_weight | Fetal weight |
| del_type | Delivery type |

novelty is the use of balanced accuracy in the performance evaluation of our models. We have brought clarity to bear on the use of this evaluation metric and assigned reasons for its use. We have also included in this research work the issue of number of different types of abortions to investigate its impact on delivery outcome decisions. This work has obtained balanced accuracy scores that are significantly high as compare to other related works in this domain. However, this work is limited in certain respects, first, is the issue of data size or sample size and population characteristics, most machine learning algorithms work best with large datasets, our work is limited in the size of data collected and detailed insight into cultural practices of sample population. Socio-economic variables such as level of education, employment status etc were excluded from the features used. Table 4 confirms variable features used and their corresponding descriptions. Label descriptions were taken from administered partographs per each patient admitted for labor.

Research limitations also includes the non-inclusion of patient body mass index (BMI) in the features collected. Observations made and conclusions drawn from electronic health records without personal interactions with patients is another limitation that would have helped clarified certain issues of concern for a contextual understanding.

## Mitigating measures

One of the mitigating measures considered was the issue of sample size. This is addressed by the use of traditional machine learning algorithms that work best with small sample size for efficient results. It is in this light that we used Logistic Regression together with Random Forest and an ensemble model Gradient Boosting classifier for comparative analysis of performance.

## Conclusions, recommendations and future work

We have shown in this research how related study results are connected to the maternal interaction model shown in Fig 1. We have also shown the effect of including history of various types of abortions as an input variable and established variable correlations between the input

variables and the output. Our predictive features with outcome correlations have shown to be subjects of related research works which confirms our result validity. We have achieved prediction accuracy scores that are comparable to related research works and even much better when compared with the use of prediction accuracies instead of balanced accuracies within this domain. It is our determination to gather large volumes of data for further predictive modeling in this regard.

## Acknowledgments

Appreciation goes to the management and staff of Kwahu Government Hospital for their immense support and assistance especially the Medical Director Dr. Kobina Awotwe Wiredu and the Clinical coordinator Dr. Nana Osei. It is worthy to express our appreciation to the Midwives at the Labour ward especially Margaret Nketia for her assistance.

## Author Contributions

**Conceptualization:** Michael Owusu-Adjei.

**Formal analysis:** Michael Owusu-Adjei, Twum Frimpong.

**Funding acquisition:** Michael Owusu-Adjei.

**Investigation:** Michael Owusu-Adjei.

**Methodology:** Twum Frimpong.

**Supervision:** James Ben Hayfron-Acquah.

**Validation:** Abdul-Salaam Gaddafi.

**Visualization:** Abdul-Salaam Gaddafi.

**Writing – review & editing:** Twum Frimpong.

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
