## [Decision Letter · Decision Letter 0]

23 Sep 2024

PDIG-D-24-00224

An AI-based approach to predict delivery outcome based on measurable factors of pregnant mothers.

PLOS Digital Health

Dear Dr. Michael Owusu-Adjei,

Thank you for submitting your manuscript to PLOS Digital Health. After careful consideration, we feel that it has merit but does not fully meet PLOS Digital Health's publication criteria as it currently stands. Therefore, we invite you to submit a revised version of the manuscript that addresses the points raised during the review process.

Please submit your revised manuscript within 60 days Nov 22 2024 11:59PM. If you will need more time than this to complete your revisions, please reply to this message or contact the journal office at digitalhealth@plos.org. Please include the following items when submitting your revised manuscript:

We look forward to receiving your revised manuscript.

Kind regards,

Cleva Villanueva, M.D., Ph.D.

Guest Editor

PLOS Digital Health

Cleva Villanueva

Guest Editor

PLOS Digital Health

Journal Requirements:

Additional Editor Comments (if provided):

The manuscript should comply with the PLOS Digital Health author guidelines. The language should be revised to ensure greater clarity and readability. Please include details about informed consent and the ethical approval granted by the corresponding committee. Since the study involves machine learning and includes variables that require quantitative analysis, be sure to incorporate the appropriate quantitative methodology.

In the limitations section, please note that the study was conducted in a single healthcare facility, which may limit the generalizability of the findings. Additionally, include a description of the neighborhood's characteristics to provide context on the socioeconomic conditions of the study population

Reviewers' comments:

Reviewer's Responses to Questions

**Comments to the Author**

1. Does this manuscript meet PLOS Digital Health’s publication criteria? Is the manuscript technically sound, and do the data support the conclusions? The manuscript must describe methodologically and ethically rigorous research with conclusions that are appropriately drawn based on the data presented.

Reviewer #1: Yes

Reviewer #2: Partly

2. Has the statistical analysis been performed appropriately and rigorously?

Reviewer #1: I don't know

Reviewer #2: No

3. Have the authors made all data underlying the findings in their manuscript fully available (please refer to the Data Availability Statement at the start of the manuscript PDF file)?

Reviewer #1: Yes

Reviewer #2: No

4. Is the manuscript presented in an intelligible fashion and written in standard English?

Reviewer #1: No

Reviewer #2: Yes

5. Review Comments to the Author

Reviewer #1: Greetings

That sounds a valuable research with interesting findings. Anyhow, i believe the manuscript hasnt been written adhering to the journal`s guidelines; some parts of the manuscript are too lengthy which decreases the readability of the paper. Moreover, it seems there is a need for a precise proof reading of the manuscript in terms of wording and grammar. While the study mentions ethical approval, it does not provide detailed information on how consent was obtained or how ethical considerations were addressed in data handling and analysis. While, the data was collected from a single healthcare facility, which may introduce biases and limit the applicability of the findings to other settings or populations, i reccomend the authors to discuss the socioeconomic and cultural attributes of the population living around the healthcare center in a paragraph or two. 

With kind regards

Reviewer #2: The qualitative method mentioned in the manuscript does not seem appropriate, given that the research uses statistical methods. The study primarily relies on machine learning models (Logistic Regression, Random Forest, and Gradient Boosting) to predict delivery outcomes, which are quantitative in nature. These statistical techniques require numerical data and focus on measurable features, like gestational age, maternal blood pressure, and cervical dilation, to make predictions. The research methodology should emphasize quantitative analysis and statistical methods, as those are the primary techniques employed. The mention of qualitative methods appears to be a misalignment.

6. PLOS authors have the option to publish the peer review history of their article (what does this mean?). If published, this will include your full peer review and any attached files.

**Do you want your identity to be public for this peer review?** For information about this choice, including consent withdrawal, please see our Privacy Policy.

Reviewer #1: Yes: Mohsen Khosravi

Reviewer #2: Yes: Marlon Machal

---

## [Editor Report · Decision Letter 1]

29 Oct 2024

PDIG-D-24-00224R1

An AI-based approach to predict delivery outcome based on measurable factors of pregnant mothers.

PLOS Digital Health

Dear Dr. Michael Owusu-Adjei,

Thank you for submitting your manuscript to PLOS Digital Health. After careful consideration, we feel that it has merit but does not fully meet PLOS Digital Health's publication criteria as it currently stands. Therefore, we invite you to submit a revised version of the manuscript that addresses the points raised during the review process.

Please submit your revised manuscript within 60 days Dec 28 2024 11:59PM. If you will need more time than this to complete your revisions, please reply to this message or contact the journal office at digitalhealth@plos.org. Please include the following items when submitting your revised manuscript:

We look forward to receiving your revised manuscript.

Kind regards,

Cleva Villanueva, M.D., Ph.D.

Guest Editor

PLOS Digital Health

Cleva Villanueva

Guest Editor

PLOS Digital Health

Additional Editor Comments (if provided):

The reviewers' comments and questions were not appropriately addressed. Table I is a crucial part of the study for understanding the analysis and results. It is titled "Statistical Distribution of Sample Population Counts." The authors appear to have studied women with one or more pregnancies, including those who may or may not have had abortions or stillbirths. However, it is unclear if women with births but no abortions were also included. This lack of clarity makes it difficult to interpret the data.

A single woman could be included in multiple groups; for example, one woman might be in the group of patients with abortions, as well as in the group of abortions with no live births, or abortions with live births. The authors need to revise this table to clarify which groups were compared to obtain the chi-square results they present. As it stands, it is impossible to determine what comparisons were made.

Additionally, ages should be expressed as the mean years ± standard error of the mean.

The manuscript cannot be published in its current form as it does not comply with the PLOS Digital Health author guidelines.
---

## [Decision Letter · Decision Letter 2]

8 Dec 2024

An AI-based approach to predict delivery outcome based on measurable factors of pregnant mothers.

PDIG-D-24-00224R2

Dear Dr. Michael Owusu-Adjei,

We are pleased to inform you that your manuscript 'An AI-based approach to predict delivery outcome based on measurable factors of pregnant mothers.' has been provisionally accepted for publication in PLOS Digital Health.

Best regards,

Cleva Villanueva, M.D., Ph.D.

Guest Editor

PLOS Digital Health

**Additional Editor Comments (if provided):**

The authors have adequately addressed all the questions and comments raised by the reviewers and the manuscript is suitalbe for publication at PLOS Digital Health

**Reviewer Comments (if any, and for reference):**

Reviewer's Responses to Questions

**Comments to the Author**

1. If the authors have adequately addressed your comments raised in a previous round of review and you feel that this manuscript is now acceptable for publication, you may indicate that here to bypass the “Comments to the Author” section, enter your conflict of interest statement in the “Confidential to Editor” section, and submit your "Accept" recommendation.

Reviewer #2: All comments have been addressed

2. Does this manuscript meet PLOS Digital Health’s publication criteria? Is the manuscript technically sound, and do the data support the conclusions? The manuscript must describe methodologically and ethically rigorous research with conclusions that are appropriately drawn based on the data presented.

Reviewer #2: Yes

3. Has the statistical analysis been performed appropriately and rigorously?

Reviewer #2: Yes

4. Have the authors made all data underlying the findings in their manuscript fully available (please refer to the Data Availability Statement at the start of the manuscript PDF file)?

Reviewer #2: Yes

5. Is the manuscript presented in an intelligible fashion and written in standard English?

Reviewer #2: Yes

6. Review Comments to the Author

Reviewer #2: (No Response)

7. PLOS authors have the option to publish the peer review history of their article (what does this mean?). If published, this will include your full peer review and any attached files.

**Do you want your identity to be public for this peer review?** For information about this choice, including consent withdrawal, please see our Privacy Policy.

Reviewer #2: **Yes: **Marlon Luca Machal
